# Molecular basis for glycan recognition and reaction priming of eukaryotic oligosaccharyltransferase

Ana S. Ramírez [1], Mario de Capitani [2], Giorgio Pesciullesi [2], Julia Kowal [1], Joël S. Bloch[1], Rossitza N. Irobalieva [1], Jean-Louis Reymond[2], Markus Aebi[3] & Kaspar P. Locher [1] ✉

Oligosaccharyltransferase (OST) is the central enzyme of $N$-linked protein glycosylation. It catalyzes the transfer of a pre-assembled glycan, $GlcNAc_2$-$Man_9Glc_3$, from a dolichyl-pyrophosphate donor to acceptor sites in secretory proteins in the lumen of the endoplasmic reticulum. Precise recognition of the fully assembled glycan by OST is essential for the subsequent quality control steps of glycoprotein biosynthesis. However, the molecular basis of the OST-donor glycan interaction is unknown. Here we present cryo-EM structures of *S. cerevisiae* OST in distinct functional states. Our findings reveal that the terminal glucoses ($Glc_3$) of a chemo-enzymatically generated donor glycan analog bind to a pocket formed by the non-catalytic subunits WBP1 and OST2. We further find that binding either donor or acceptor substrate leads to distinct primed states of OST, where subsequent binding of the other substrate triggers conformational changes required for catalysis. This alternate priming allows OST to efficiently process closely spaced $N$-glycosylation sites.

$N$-glycosylation is an essential post-translational modification in eukaryotic cells. $N$-glycans attached to newly synthesized proteins play a critical role in protein folding, export, and endoplasmic reticulum (ER) homeostasis[1–3]. The process starts with the biosynthesis of a lipid-linked oligosaccharide (LLO), which involves the sequential action of multiple glycosyltransferases, encoded by *ALG* (asparagine-linked-glycosylation) genes[4–13] (Supplementary Fig. 1a). In metazoans, plants, and fungi, the assembled LLO is $DolPP-GlcNAc_2Man_9Glc_3$, where DolPP refers to dolichylpyrophosphate, GlcNAc to N-acetylglucosamine, Man to mannose, and Glc to glucose. In contrast to higher eukaryotes, certain unicellular eukaryotes produce shorter oligosaccharides due to the absence of specific *ALG* genes in their genomes[14–16]. The transfer of the pre-assembled glycan onto secretory proteins is catalyzed by oligosaccharyltransferase (OST), provided these proteins contain an $N$-glycosylation sequon N-X-S/T[1–3,17,18]. Following glycan transfer, glucosidase-I and glucosidase-II sequentially remove the two terminal glucose units of the newly attached glycan. The resulting $GlcNAc_2Man_9Glc$ structure is recognized by the ER-luminal lectins calreticulin (CRT) and calnexin (CNX), which recruit specific chaperones to direct the glycoproteins either to folding or to degradation[2].

In eukaryotes, the last steps of LLO assembly and the glycosylation reaction catalyzed by OST occur in the lumen of the ER. This leads to a competition of incomplete LLO intermediates and fully assembled $DolPP-GlcNAc_2Man_9Glc_3$ as donor substrates for OST[3]. Both in vivo and in vitro, OST has been shown to exhibit high specificity towards the fully assembled glycan containing a terminal $Glc_3$ moiety[8,9,19–21], whereas intermediates are only used as donor substrates when accumulated in the cell due to defects in LLO assembly[22–25]. Two effects could in principle explain the preference of eukaryotic OST towards fully assembled LLO: First, DolPP-linked glycan intermediates might be present in such low amounts in the cell that their transfer is negligible. Second, OST might specifically bind the fully assembled, $Glc_3$-containing oligosaccharide. The first hypothesis is incompatible with the experimental finding that biosynthetic LLO intermediates amount to

[1]Institute of Molecular Biology and Biophysics, Eidgenössische Technische Hochschule (ETH), Zürich, Switzerland. [2]Department of Chemistry and Biochemistry, University of Bern, Bern, Switzerland. [3]Institute of Microbiology, Eidgenössische Technische Hochschule (ETH), Zürich, Switzerland. ✉e-mail: locher@mol.biol.ethz.ch

~30% of all DolPP-containing glycans in bovine pancreas[26]. How OST differentiates between biosynthetic intermediates and full-length LLO remains unknown.

In higher metazoans and fungi, OST is a multimeric complex composed of eight subunits spatially arranged in three subcomplexes that are formed during OST assembly[27]. Subcomplex I contains OST1 and OST5; subcomplex II contains OST3 (or OST6), STT3, and OST4; and subcomplex III contains OST2, WBP1, and SWP1. The principal catalytic subunit STT3 and the oxidoreductases OST3 and OST6 display enzymatic activity, and the subcomplex II has therefore a well-defined role within OST. In contrast, the role of the other subcomplexes has remained unclear. Recent cryo-EM structures provided insight into the architecture of yeast and human OSTs and outlined pockets that could interact with substrates[28–31]. Particularly, the structure of human OST-B contained co-purified dolichyl-phosphate (DolP) and an unidentified acceptor substrate[30]. However, the recognition of fully assembled glycan is unknown. Here we present high-resolution structures of *S. cerevisiae* OST (yeast OST) bound to chemically well-defined substrate analogs, including a chemo-enzymatically generated LLO carrying the complete oligosaccharide $GlcNAc_2Man_9Glc_3$. Our findings revealed that LLO binding involves recognition of the $GlcNAc_2$ and $Glc_3$ moieties by non-catalytic subunits of OST. We further found that OST can be primed by either donor or acceptor binding, but only the binding of both substrates induces the conformational changes in the catalytic STT3 subunit that are required for catalysis.

## Results

### Chemo-enzymatic generation of Dol20-PP-GlcNAc₂Man₉Glc₃

The complex structure of the LLO substrate of eukaryotic OST represents a major hurdle for in vitro functional and structural studies. On the one hand, the dolichyl tail is between fourteen and twenty-one isoprenyl units long[32–34], making it highly insoluble in aqueous solutions. On the other hand, it contains a large and branched oligosaccharide, which would require highly elaborated chemical synthesis[35,36]. To overcome these obstacles, we reconstituted the complete biosynthetic pathway generating $DolPP-GlcNAc_2Man_9Glc_3$ in vitro (Supplementary Fig. 1a), using as the initial substrate the chemically synthesized precursor Dol20-PP-$GlcNAc_2$, which contains the first four isoprenyl units of the native dolichyl lipid and is soluble in water (Supplementary Fig. 1b)[37]. Our approach involved the heterologous expression and purification of the ALG enzymes required for chemo-enzymatic glycan elongation and the chemical synthesis of the donor substrates Dol25-P-Man and Dol25-P-Glc required for the last seven steps of the biosynthesis[7]. We first produced Dol20-PP-$GlcNAc_2Man_5$ using ALG1, ALG2, and ALG11, all of which utilize GDP-Man as a donor substrate[38]. We subsequently performed the addition of four mannose units catalyzed by ALG3, ALG9, and ALG12, using Dol25-P-Man as donor substrate. This yielded Dol20-PP-$GlcNAc_2Man_9$, where the so-called B- and C-branches are completed[39]. We finally added three glucose units from Dol25-P-Glc to branch A using the purified glucosyltransferases ALG6, ALG8, and ALG10. This resulted in the final product Dol20-PP-$GlcNAc_2Man_9Glc_3$ (Figs. 1a, b).

We verified the correct assembly of Dol20-PP-$GlcNAc_2Man_9Glc_3$ by using it as a donor substrate for transfer onto the fluorescently labeled peptide TAMRA-YANATS-NH₂ using purified yeast OST complex[40]. SDS-PAGE-tricine gels analysis showed a change in the mass of the produced glycopeptide compared to the starting Dol20-PP-$GlcNAc_2$ (Fig. 1c). Furthermore, liquid chromatography-tandem mass spectrometry (LC-MS/MS) analysis of the glycopeptides revealed that the major ion peak *m/z* values for the elongated glycan were consistent with the addition of twelve hexoses (Fig. 1c), confirming the correct assembly of Dol20-PP-$GlcNAc_2Man_9Glc_3$.

## Dolichylpyrophosphate binding and glycan recognition by OST

We performed single-particle cryo-EM analysis of a sample containing nanodisc-reconstituted yeast OST, MnCl₂, chemo-enzymatically generated Dol20-PP-GlcNAc₂Man₉Glc₃, and the non-reactive peptide TAMRA-YA(Dab)ATS-NH₂, where the acceptor asparagine was replaced by 2,4-diaminobutanoic acid (Dab)[31,41–43] (Fig. 1d). Dab-containing peptides have been successfully used as competitive inhibitors of bacterial and eukaryotic OST enzymes[41–43], although they bind with lower affinity than reactive peptides[31,42]. We used the fluorescently labeled peptide TAMRA-YA(Dab)ATS-NH₂, which exhibits stronger inhibition than its unlabeled counterpart YA(Dab)ATS-NH₂, suggesting that the TAMRA entity increased the binding affinity[31]. We identified two distinct functional states of OST in this sample (Supplementary Fig. 2): A binary complex containing bound LLO (resolved at 3.0 Å resolution) and a ternary complex containing both donor and acceptor substrates (resolved at 3.1 Å resolution) (Fig. 1d and Supplementary Table 1).

In the structure containing Dol20-PP-GlcNAc₂Man₉Glc₃, we observed well-defined EM density covering the catalytically important divalent ion Mn²⁺, the four isoprenyl units of Dol20, the pyrophosphate group, and the entire A-branch glycan, which includes two GlcNAc units at the reducing end, four mannose units, and the three terminal glucoses (Fig. 1d and Supplementary Fig. 3). The Dol20 lipid reaches approximately halfway across the membrane and is accommodated in a hydrophobic groove formed by TM6 and TM11 of the catalytic subunit STT3 (Fig. 2a, Supplementary Fig. 4). The pyrophosphate group is coordinated by the manganese (II) ion and the STT3 residues Trp208 and Arg404 (Supplementary Fig. 5). The A-branch of the glycan revealed a stretched or extended conformation, oriented parallel to the membrane plane, and is in contact with OST residues over its entire length. While all the nine hexose units of the A-branch were well defined in the EM density, we identified two regions where the glycan was specifically recognized: (1) the GlcNAc₂ group at the reducing-end and (2) the terminal Glc₃ moiety at the non-reducing end (Fig. 2a). In contrast, the mannoses of the A-branch are forming weak van der Waals contacts with OST. The recognition of GlcNAc₂ exclusively involves residues of the catalytic subunit STT3. Close to the active site, Tyr521 and Asn536 form hydrogen bonds with the N-acetyl group of the first GlcNAc, while Thr537 interacts with the second GlcNAc unit (Fig. 2b). At the non-reducing end, the terminal Glc₃ are accommodated in a binding pocket formed by the auxiliary subunits OST2 and WBP1. Specifically, the terminal α,1-2 glucose interacts with Tyr383 of the WBP1 subunit via CH-π stacking, whereas the penultimate glucose interacts via hydrogen bonding with Asn75 of the OST2 subunit (Fig. 2c). The pocket accommodating the three glucose units is already formed in the apo-state of OST (Supplementary Fig. 6) and no conformational changes are required to accomplish shape complementarity.

In contrast to the A-branch, the B- and C-branches of bound LLO only show weak EM density, suggesting that they are more dynamic (Supplementary Fig. 3). Our findings imply that glycan recognition by OST mainly depends on the integrity of the A-branch.

### The terminal Glc₃ moiety in the donor substrate increases OST efficiency in vitro

To assess the functional impact of LLO glycan completeness on OST activity, we performed in vitro glycosylation assays using chemo-enzymatically generated Dol20-PP-GlcNAc₂Man₉Glc₃ and selected intermediates (Fig. 2d). We observed the highest turnover rate for the LLO carrying the complete glycan GlcNAc₂Man₉Glc₃, whereas all intermediates tested exhibited reduced glycosylation rates (Fig. 2d). The most prominent reduction (by ~75%) was observed for Dol20-PP-GlcNAc₂Man₉, whereas the earlier intermediates containing GlcNAc₂-Man₅ or GlcNAc₂ showed a 50% decrease in the glycosylation rate when compared to the Glc₃-containing glycan. The even shorter Dol20-PP-

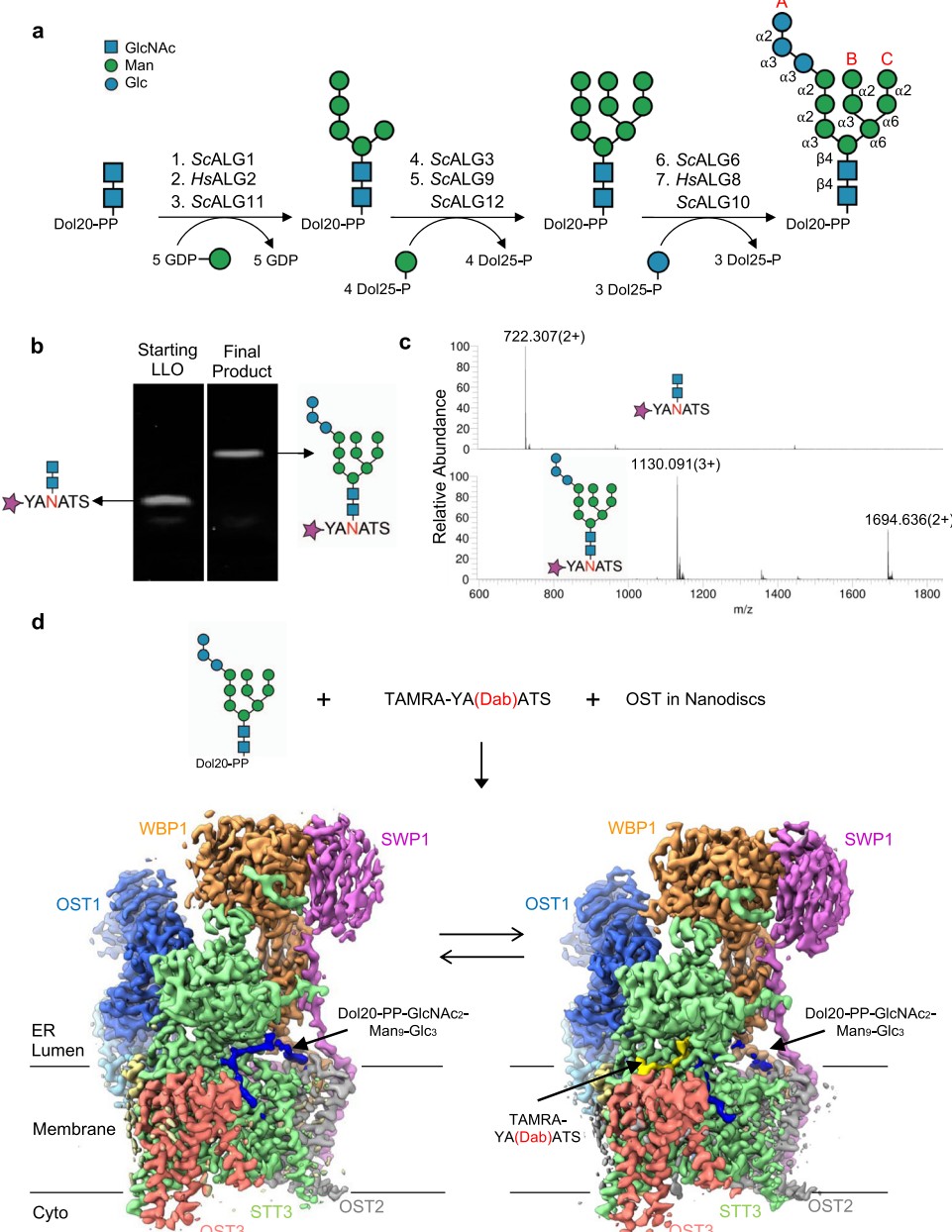

**Fig. 1 | Donor LLO generation and cryo-EM structures of yeast OST with bound substrates. a** Schematic of the chemo-enzymatic synthesis of Dol20-PP-GlcNAc$_2$Man$_9$Glc$_3$. Branches A, B, and C are labeled in the final product and glycosidic linkages are indicated. Enzyme names above reaction arrows denote purified ALG proteins. **b** Representative tricine-SDS-PAGE analysis of fluorescently lapeled peptide TAMRA-YANATS-NH$_2$ glycosylated in vitro using purified yeast OST and either Dol20-PP-GlcNAc$_2$ or the final product Dol20-PP-GlcNAc$_2$Man$_9$Glc$_3$.

$n > 3$. **c** LC-MS/MS analysis of the glycopeptides shown in **b**. Theoretical masses for glycopeptide carrying GlcNAc$_2$: $z = 2$, 722.6, and for glycopeptide carrying GlcNAc$_2$Man$_9$Glc$_3$: $z = 2$, 1694.6/$z = 3$, 1129.7. **d** Cryo-EM analysis of a sample containing OST, Dol20-PP-GlcNAc$_2$Man$_9$Glc$_3$, and TAMRA-YA(Dab)ATS-NH2 peptide yielded two distinct states presumably in equilibrium: LLO-bound (left) and ternary complex (right). Cryo-EM maps for both states are shown and labeled.

GlcNAc was not accepted as a substrate by yeast OST, which is in line with earlier studies performed with yeast microsomes membranes[44]. Our biochemical results support our structural observations and previous genetic evidence that suggest important functionality of the two GlcNAc moieties at the reducing end and the terminal glucoses on LLO recognition.

## Structure of a ternary complex of OST with bound substrates

The generation of the substrate analogs TAMRA-YA(Dab)ATS and Dol20-PP-GlcNAc$_2$Man$_9$Glc$_3$ enabled us to visualize a ternary complex of eukaryotic OST, and well-defined EM density covering bound peptide, which allowed us to build the complete hexapeptide chain

(Figs. 3a, 3b). Intriguingly, clear density was also observed for the attached TAMRA fluorophore in a pocket formed between the subunits STT3 and OST3, a location that would accommodate the nascent polypeptide chain N-terminal to the glycosylation sequon (Supplementary Fig. 7). The binding of TAMRA to OST explains the higher affinity observed for TAMRA-YA(Dab)ATS compared to its unlabeled counterpart.

In the ternary complex, the external loop EL5 of the STT3 subunit is ordered and packs against the luminal domain of STT3, resulting in the formation of a tunnel that connects the binding sites of acceptor and donor substrates[18,30,43,45,46] (Fig. 3c). The side chain of the Dab residue is lodged in this tunnel (Fig. 3c), with its terminal amino group

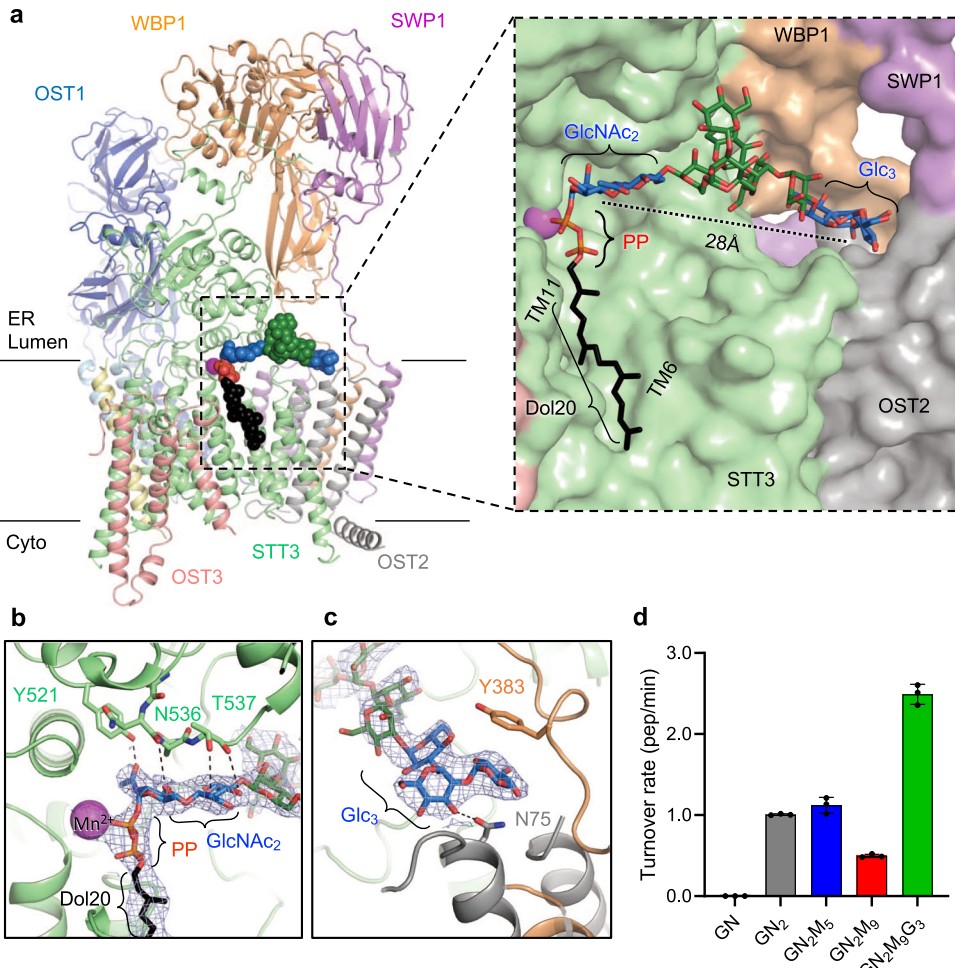

**Fig. 2 | Structure of *S. cerevisiae* OST with bound LLO analog. a** OST structure is shown in cartoon representation. Subunits are colored individually and labeled. Bound Dol20-PP-GlcNAc$_2$Man$_9$Glc$_3$ is shown as spheres, with carbons of the Dol20 colored black and those of the glucose and *N*-acetylglucosamine units colored blue and of the mannose units colored green. The inset shows a close-up view of the LLO binding site, with OST shown as surface and subunits colored as in the ribbon diagram and labeled. Dashed line indicates the distance between the recognition sites of GlcNAc$_2$ and of Glc$_3$. **b** Close-up view of the interactions of the STT3 subunit with the GlcNAc$_2$ entity. Glycan units are shown as sticks and colored as in **a**. Residues involved in binding are shown as sticks and labeled. **c** Close-up view of the interactions of OST with the Glc$_3$ moiety. Glycan units are shown as sticks and colored as in **a** OST2 is colored gray, WBP1 is colored orange. Residues involved in binding are shown as sticks and labeled. EM densities are shown as a blue mesh. **d** In vitro glycosylation analysis of purified yeast OST with LLO analogs carrying different glycan moieties. Data are presented as mean values ± SD ($n = 3$). Individual data points are depicted as dots. Source data provided as source file.

positioned at a distance of 3.3 Å from the C-1 of the reducing-end GlcNAc of the glycan donor. This suggests that our structure reflects a state preceding the nucleophilic attack on the reducing end GlcNAc, akin to a pseudo-Michaelis complex. In this conformation, two residues from the EL5 contribute and complete the active site of the enzyme: Glu350 and Arg328. The first of these (Glu350) has been shown to be catalytically important as it contributes to the deprotonation of the amide group of the acceptor asparagine, along with Asp47[39,43]. The second, Arg328, provides a salt bridge with the pyrophosphate group of bound LLO (Supplementary Fig. 5). This interaction is not present in the binary complex of OST with bound LLO state, where the pyrophosphate group only interacts with Trp208 and Arg404. This suggests that Arg328 is involved in stabilizing the leaving group DolPP but not in DolPP-GlcNAc$_2$-Man$_9$-Glc$_3$ binding.

The manganese (II) ion is coordinated by residues of the DxD/E motif[47] (Asp166 and Glu168), the catalytic residues Asp47 and Glu350, and by the pyrophosphate group of the LLO (Fig. 3b). Interestingly, the Dab residue also interacts with the STT3 side chain of Asn535, which is part of an *N*-glycosylation sequon ($^{535}$N$^{536}$N$^{537}$T) that is not processed in vivo, possibly because an *N*-glycan attached to Asn535 would prevent peptide binding.

To distinguish the effects of bound LLO and peptide on the 3D conformation of OST, we determined a separate structure of yeast OST bound to the reactive peptide TAMRA-YANATS-NH$_2$ but in the absence of LLO (Supplementary Fig. 8, Table 1). A comparison with the ternary complex structure (containing Dab peptide) revealed indistinguishable peptide pose and conformation, suggesting that the Dab residue used for the ternary complex structure had not altered the conformation of OST (Fig. 3c). Importantly, both structures featured key contacts of the hydroxyl group of the +2 threonine with the "WWD" motif of STT3, which is essential for sequon recognition by OST[47,48] (Fig. 3c).

**Conservation of LLO glycan recognition in eukaryotic organisms**
When comparing different eukaryotic species, it is notable that a limited spectrum of *N*-glycans is transferred onto secretory proteins. These range from GlcNAc$_2$ to GlcNAc$_2$Man$_9$Glc$_3$ depending on the presence or absence of ALG enzymes in the different species[1]. Having identified STT3, WBP1, and OST2 as the *S. cerevisiae* OST subunits that provide specific interactions with bound Dol20-PP-GlcNAc$_2$Man$_9$Glc$_3$ (Fig. 4a), we performed protein sequence alignments with the equivalent subunits of other species to evaluate the conservation of

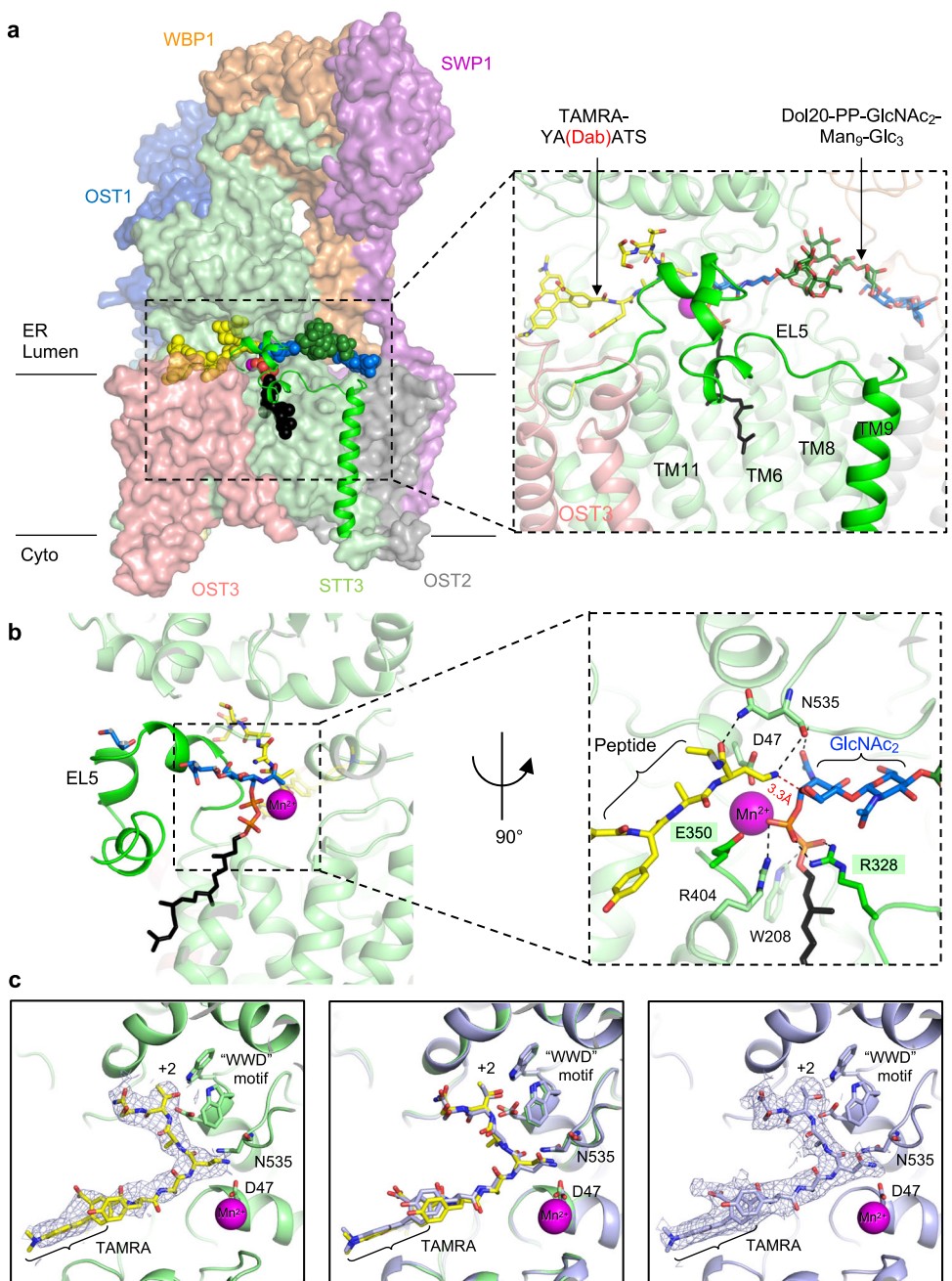

**Fig. 3 | Structure of a ternary complex of OST. a** OST is shown in surface representation. Subunits are colored and labeled. Ordered EL5 and TM9 are shown in ribbon representation. Bound substrates are shown in sphere representation. Dab-containing peptide is colored yellow. Dol20 is colored black, glucose and N-acetylglucosamine units colored blue and mannose units colored green. The inset shows a close-up view of the binding sites for acceptor peptide and LLO. Substrates are shown as sticks and labeled **b**. Active site of STT3, with inset showing a close-up view. OST is shown in ribbon representation and colored as in **a**. Substrates are shown as sticks and colored as in **a**. Residues involved in substrate binding and catalysis are shown as sticks and labeled. The dashed red line indicates the distance between the nitrogen of the amino group of the Dab residue and C1 of the reducing-end GlcNAc. Residues of EL5 involved in substrate binding are shaded green. **c** Left, binding of the Dab-containing peptide in the ternary complex. Center, superposition of OST structures with bound Dab-containing peptide in the ternary complex (yellow and green) and bound reactive peptide in the peptide-bound state (light blue). Right, binding of the reactive peptide to OST in peptide-bound state. EM densities are shown as a light blue mesh.

the relevant residues (Fig. 4b). We first analyzed the GlcNAc$_2$ binding site and found that the tyrosine residue forming a hydrogen bond with the acetamido group of the reducing-end GlcNAc (Tyr521 in *S.c.* STT3) is strictly conserved not only in eukaryotes, but also in bacteria and archaea that transfer oligosaccharides containing an N-acetylhexose at their reducing end. In addition, we found a high degree of conservation of Thr537 in STT3 homologs where the second glycan unit is GlcNAc[14] (Fig. 4b). This suggests that all these OST enzymes bind the

reducing end *N*-acetylhexoses of their LLO substrates in the same way as visualized in our structure of yeast OST. We next analyzed WBP1 and OST2 residues involved in forming the binding site of the Glc$_3$ moiety and found a high conservation of the residues interacting with the terminal α,1-2 glucoses in multimeric OSTs that transfer Glc$_3$-containing oligosaccharides (Fig. 4b). Our analysis shows that the increase in the length of the A-branch in eukaryotic LLOs correlates with the presence of the OST subunits WBP1 and OST2 that form the binding

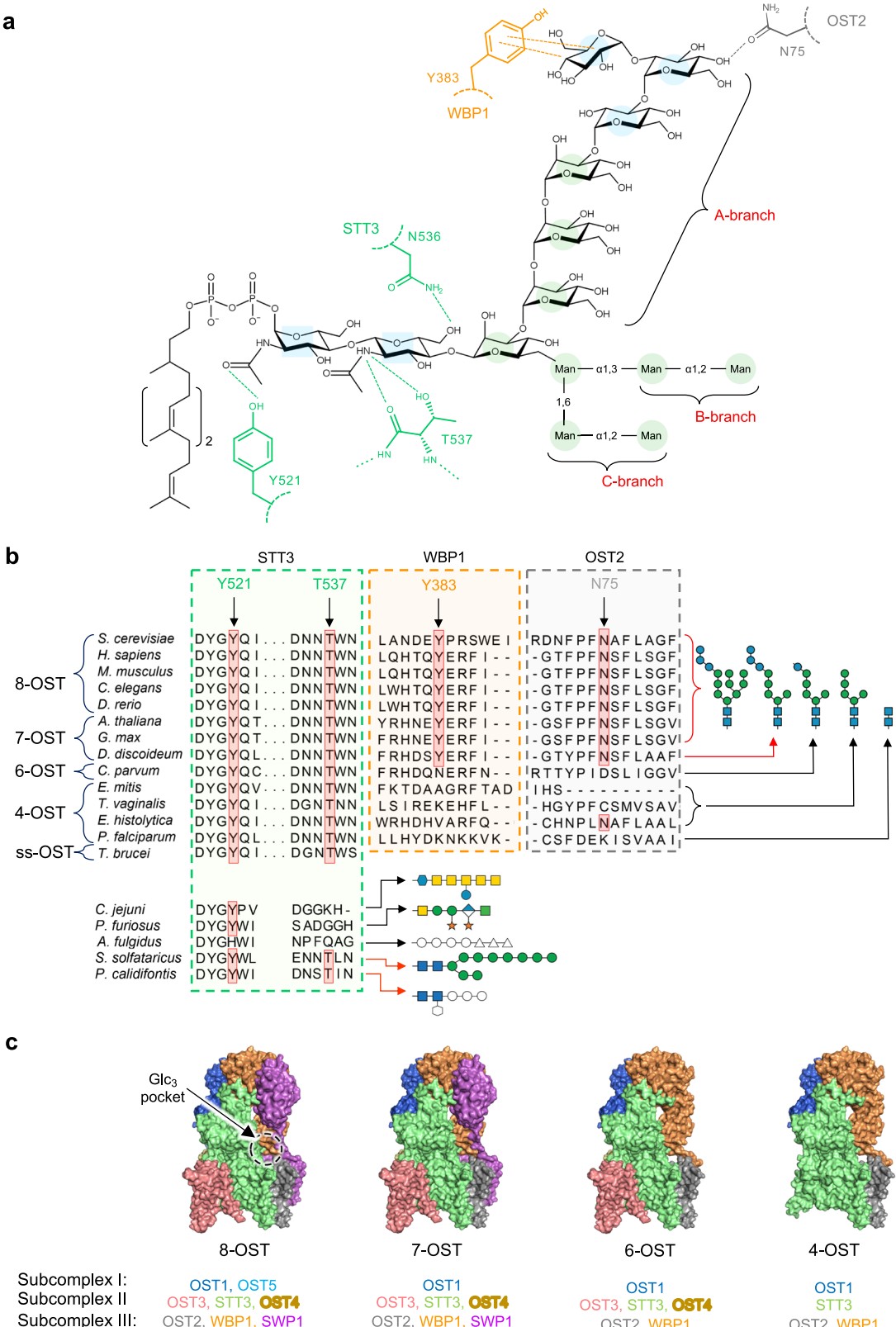

**Fig. 4 | Conservation of LLO binding motifs in OST enzymes. a** Interactions of yeast OST with GlcNAc$_2$Man$_9$Glc$_3$ shown as a structure. Side chains of STT3, OST2, and WBP1 subunits are colored green, gray, and orange, respectively, and labeled in single-letter code. Dashed lines indicate hydrogen bonds or CH2-π interactions **b** Protein alignment of OST subunits involved in glycan recognition. For STT3, residues 518–523 and 534–539 are aligned; for WBP1, residues 378–389 are aligned,

and for OST2, residues 69–81 are aligned. For *H. sapiens* and *T. brucei* the STT3A orthologs were used in the alignment. **c** Surface representation of OST models with different subunit compositions. The octameric OST (8-OST) corresponds to the structure of *S. cerevisiae* OST. For 7-OST, 6-OST, and 4-OST, the models were generated by removing the specific subunits from the yeast OST structure.

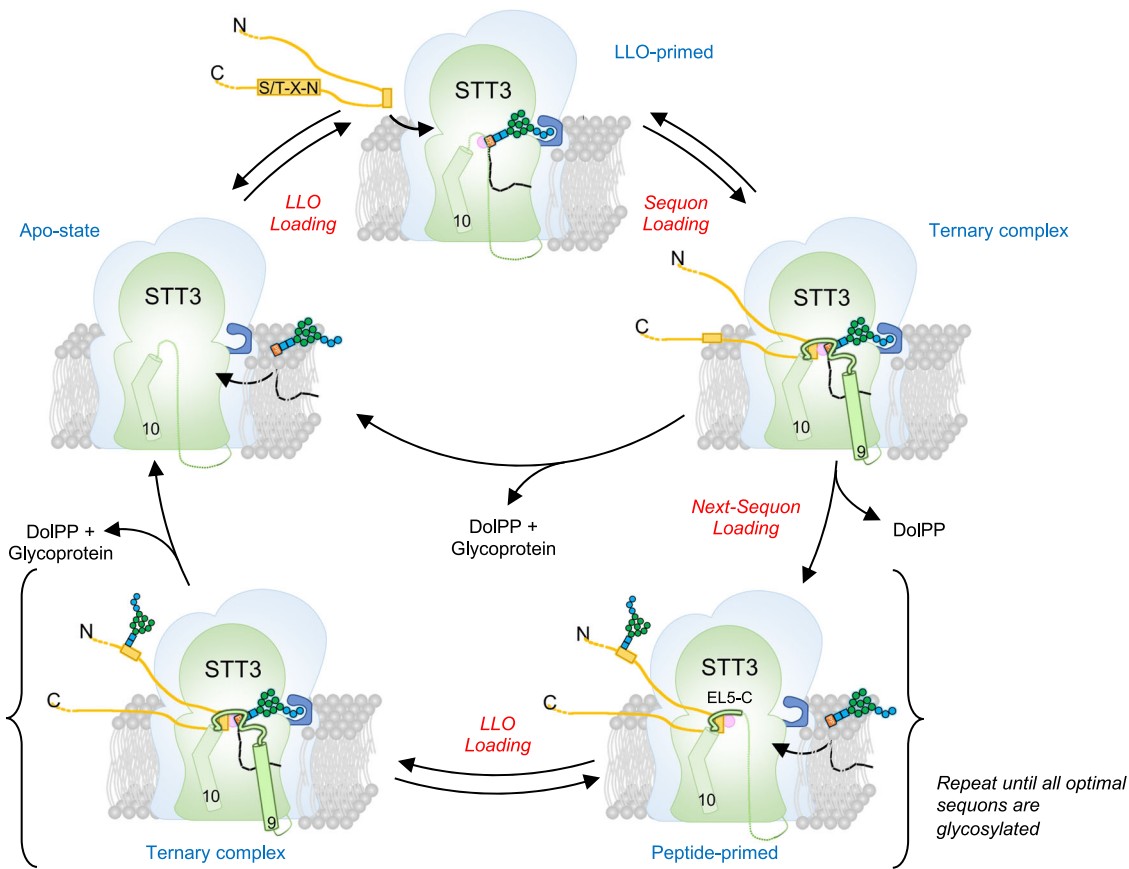

**Fig. 5 | Proposed glycosylation cycle of OST.** *Apo-state:* in the absence of bound substrates, EL5 and TM9 of STT3 are shown as a dashed line, indicating flexibility. *LLO-primed:* OST bound to donor substrate, with EL5 and TM9 remaining flexible. The terminal Glc$_3$ moiety is accommodated in the Glc$_3$-binding pocket. *Ternary complex*: binding of donor and acceptor substrates promotes engagement of EL5 and TM9, completing the active site of OST. Upon glycan transfer, the glycopeptide and dolichylpyrophosphate are released, and OST can start a new catalytic cycle. *Peptide-primed*: if the next glycosylation sequon is closely spaced, it can bind to OST, leading to engagement of the C-terminal half of EL5 (EL5-C).

pocket for the non-reducing end Glc$_3$ (Fig. 4c), suggesting a coordinated evolutionary path of *N*-linked protein glycosylation reflected in OST architecture and oligosaccharide structure.

## Discussion

The structure of LLO-bound OST provides essential insight into glycan recognition. Although the GlcNAc$_2$Man$_9$Glc$_3$ oligosaccharide could in principle form many polar interactions with OST, its specific recognition involves only the GlcNAc units at the reducing end and the glucose units at the non-reducing end of the A-branch, two regions that represent the initiating and terminating steps of LLO biosynthesis. The mannose units between these recognition motifs are ordered because the A-branch is in a stretched conformation that does not tolerate other conformers and because it is in Van der Waals distance to the STT3A surface. In contrast, the B- and C- branches of the LLO are not bound by OST and their mannose units remain flexible, which might account for the lower in vitro glycosylation efficiency of purified OST when transferring GlcNAc$_2$Man$_9$ compared to GlcNAc$_2$Man$_5$ (Fig. 2d). Since the B- and C-branches are not recognized by OST, their structural integrity must be controlled in a step prior to oligosaccharide transfer. Potential candidates to perform this quality control are the glucosyltransferases (ALG6, ALG8, or ALG10) involved in the late steps of LLO assembly. We conclude that the reason OST exclusively transfers GlcNAc$_2$Man$_9$Glc$_3$ in vivo is a consequence of (1) the bipartite binding of the oligosaccharide, and (2) the strict order of LLO assembly, which is given by the topological separation of the biosynthetic pathway and the high substrate specificity of the lumen-oriented glycosyltransferases.

Our structural and functional findings provide sufficient molecular insight to propose a modular mechanism for the OST-catalyzed reaction cycle, which involves structural rearrangements of the EL5 and TM9 regions of the catalytic subunit STT3 (Fig. 5). Binding of LLO to an apo-state of OST leads to the formation of an "LLO-primed" state, where EL5 and TM9 remain unengaged and allowing access of a glycosylation sequon to the active site (Supplementary Fig. 10). In this state, the active site of is not fully formed, which probably helps to prevent futile hydrolysis of bound LLO. Upon binding of an acceptor peptide, the complete EL5 and TM9 become ordered, resulting in a catalytically active state. Conversely, binding of an acceptor peptide alone to the apo-state engages the C-terminal half of EL5 (EL5-C) only, and does not involve TM9 (Supplementary Fig. 9). This still allows the subsequent binding of LLO that now engages the N-terminal half of EL5 (EL5-N) and TM9 (Supplementary Fig. 10). The sequential ordering of EL5 and TM9 suggest that an OST enzyme primed with a single substrate may display a higher affinity to the other substrate as compared to the apo-enzyme. We speculate that this is important for the efficient glycosylation of closely spaced sequons, where there is an increase in the local sequon concentration, and a peptide-primed OST is more likely to be formed. After glycan transfer, product release is achieved by the inability of glycopeptides to bind OST due to steric clashes with EL5 upon formation of the β1-glycosidic linkage[42]. The leaving group DolPP is stabilized by conserved residues of STT3 and is further processed by the DolPP-specific pyrophosphatase CWH8[49].

The mechanistic insight on fully assembled glycan recognition described here relied on our chemo-enzymatic approach for the preparation of soluble LLOs, which offered precise control of the

synthesized oligosaccharide structures. Our pipeline for generating LLO intermediates is therefore a powerful tool for mechanistic studies not only of OST, but also for future studies of the LLO-processing ALG enzymes, for which specific substrate recognition is essential for the ordered assembly of the growing and branched high-mannose glycan.

Multimeric OSTs are composed of three distinct subcomplexes. Whereas the catalytic function of subcomplex II (STT3, OST3, and OST4) is clearly defined, the role of the non-catalytic subcomplexes I (OST1 and OST5) and III (WPB1, OST2, and SWP1) has been unclear. Our results reveal that subcomplex III has a key functional role in recognizing fully assembled LLO by providing a binding pocket for the terminal $Glc_3$ moiety. The location of the remaining subcomplex I relative to the catalytic center, and its association with translating ribosomes in human OST-A[30,50], suggest that it might be involved in the expansion of polypeptide substrates.

In conclusion, our results describe the structural basis of LLO-glycan recognition and priming by OST in higher eukaryotes and assign functional roles to non-catalytic OST subunits. Our findings will also contribute to the design of specific OST inhibitors with pharmacological potential[51,52].

## Methods

### Expression and purification of yeast OST3-containing OST complexes

12 L of yeast culture (MAT α his3Δ1 leu2Δ0 lys2Δ0 ura3Δ0 arg4Δ0 ost6::LEU2MX6 OST4-1D4::kanMX6 YEp352-OST3)[28] were grown at 30 °C in synthetic dropout medium lacking uracil (6.7 g/l yeast nitrogen base w/o amino acids, 20 g/l glucose and 1.92 g/l SD supplement -Ura, 20 g/l glucose and 1.92 g/l SD supplement -Ura) using 5 L baffled glass flasks to an optical density ($OD_{600\,nm}$) between 3.0 and 4.0. The cells were harvested by centrifugation at $5500 \times g$. and the cell pellets were flash-frozen in liquid nitrogen and stored at −80 °C until use.

For membrane preparation, cells were resuspended in ice-cold lysis buffer containing 50 mM HEPES, pH 7.5, 200 mM NaCl, and 1 mM $MgCl_2$, supplemented with protease inhibitors (EDTA-free protease inhibitor cocktail (Roche), 2.6 mg/l aprotinin, 5 mg/l leupeptin, 1 mg/l pepstatin, 2 mM benzamidine HCl). The cells were lysed in a BioSpec Beadbeater with 1 min/5 min, on/off cycles (4 cycles) in a water-ice bath, and all following steps were performed at 4 °C. Unbroken cells were removed by centrifugation for 10 min at $3000 \times g$. Membranes were collected by ultracentrifugation at $150,000 \times g$ for 1 h, resuspended in membrane solubilization buffer containing 50 mM HEPES, pH 7.5, 500 mM NaCl, 1 mM $MgCl_2$, 1 mM $MnCl_2$ 10% (v/v) glycerol, and subsequently incubated with a mixture of 1% (w/v) $N$-dodecyl-β-D-maltopyranoside (DDM, Anatrace) and 0.2% (w/v) cholesteryl hemisuccinate (CHS, Anatrace) for 1.5 hours. The supernatant was collected after high-speed centrifugation at $150,000 \times g$ for 30 min and incubated with Sepharose-coupled Rho-1D4 antibody (University of British Columbia) for three hours. The beads were washed with solubilization buffer containing 0.03% (w/v) DDM and 0.006% (w/v) CHS and subsequently with washing buffer containing 50 mM HEPES, pH 7.5, 250 mM NaCl, 1 mM $MgCl_2$, 1 mM $MnCl_2$ 10% (v/v) glycerol, 0.03% (w/v) DDM, and 0.006% (w/v) CHS. OST complex was eluted with 0.5 mg/mL 1D4 peptide (GenScript) in washing buffer, concentrated, and further purified by size exclusion chromatography using a Superose 6 column (GE Life Sciences) with a buffer containing 50 mM HEPES, pH 7.5; 150 mM NaCl, 1 mM $MgCl_2$, 1 mM $MnCl_2$, 5% (v/v) glycerol, 0.03% (w/v) DDM and 0.006% (w/v) CHS. The peak fractions were collected and used for nanodisc-reconstitution.

For structural studies performed with OST complex in digitonin, the protocol was changed slightly. After immobilization of OST complexes in Sepharose-coupled Rho-1D4 antibody, the beads were washed with washing buffer containing 0.03% (w/v) DDM and 0.006% (w/v) CHS, followed by washing buffer containing 0.03% (w/v) DDM and 0.006% (w/v) CHS and 0.1% digitonin (Merck), and a last step with washing buffer containing only 0.1% w/w digitonin. OST complex was eluted with 0.5 mg/mL 1D4 peptide (GenScript) in washing buffer containing 0.1% w/w digitonin, concentrated, and further purified by size exclusion chromatography using a Superose 6 column (GE Life Sciences) with a buffer containing 25 mM HEPES, pH 7.5; 150 mM NaCl, 1 mM $MnCl_2$, 0.1% digitonin. The peak fractions were collected, concentrated, and used for the preparation of cryo-EM grids.

### Nanodisc-reconstitution of yeast OST3-containing OST complexes

A mixture of *E. coli* polar lipids and L-α-phosphatidilcholine (Avanti Polar Lipids) was solubilized in 0.5% DDM. The membrane scaffold protein (MSP) 1D1, was expressed and purified as described previously. Purified OST complex was reconstituted in a mixture of 1:10:350 (OST:MSP:Lipids), as following. First, OST complex was mixed with the solubilized lipids for 10 minutes at room temperature, subsequently MSP was added and the mixture was incubated for additional 30 min. The detergent was removed by addition activated Biobeads (Bio-Rad) and incubation at 4 °C overnight. After removal of the Biobeads, the reconstitution mixture was incubated with Sepharose-coupled Rho-1D4 antibody for 2 h. The beads were washed with buffer containing 50 mM HEPES, pH 7.5; 150 mM NaCl, 1 mM $MgCl_2$, and 1 mM $MnCl_2$. Nanodisc-reconstituted OST complex was eluted with 0.5 mg/mL 1D4 peptide in washing buffer, concentrated, and further purified by size exclusion chromatography using a Superose 6 column with a buffer containing 25 mM HEPES, pH 7.5; 150 mM NaCl and 5 mM $MnCl_2$. The peak fractions were collected and used either for cryo-EM studies or in vitro glycosylation assays.

### Purification of ALG enzymes

Purification of ALG1, ALG2, ALG11, ALG3, ALG9, ALG12 and ALG6 was performed as described previously[38,39].

The *ALG8* gene from *H. sapiens* was amplified using cDNA from human embryonic kidney cells (HEK293T) as template, and cloned into a modified pUC57 vector carrying an eYFP-1D4 tag at the C-terminus. ALG8 construct was expressed by transient transfection in HEK293T cells, using branched polyethyleneimine (PEI) in a 2:1 (w/w) PEI:DNA ratio and harvested after 24 hours. For protein purification, cells were resuspended in lysis buffer containing 50 mM HEPES, pH 7.5; 250 mM NaCl; 10% (w/v) glycerol, supplemented with protease inhibitors (EDTA-free protease inhibitor cocktail (Roche), 2.6 mg/l aprotinin, 5 mg/l leupeptin, 1 mg/l pepstatin, 2 mM benzamidine HCl). Lysis was performed by dounce homogenization, followed by solubilization with 1% (w/v) DDM, 0.2% (w/v) CHS, for 1 hour. The supernatant was collected after high-speed centrifugation at $150,000 \times g$ for 30 min and incubated with Sepharose-coupled Rho-1D4 antibody (University of British Columbia) for 1 hour. The beads were washed with washing buffer containing 25 mM HEPES pH 7.4, 150 mM NaCl, 10% glycerol and 0.03% DDM, and 0.006% CHS. ALG8 was eluted with 0.2 mg/mL 1D4 peptide (GenScript) in washing buffer for 1 hour and concentrated to a concentration of ~0.5 mg/mL.

A codon-optimized synthetic gene (GenScript) of *ALG10* from *S. cerevisiae* was fused to an N-terminal Flag-eYFP-3C tag and cloned into a pOET1 vector (Oxford Expression Technologies). Baculovirus production was performed using *flashBAC* DNA (Oxford Expression Technologies) in *Spodoptera frugiperda* (Sf9) cells following the manufacturer's instructions. Cells were grown in serum-free SF900 medium, infected at a density of $1 \times 10^6$ cells/mL, and harvested after 72 hours. For purification, the cells were resuspended in a buffer containing 50 mM NaCl, 50 mM HEPES pH 7.4, supplemented with 0.1 mg/mL DNAse I, 1% (v/v) protease inhibitor cocktail (Sigma), and 0.1 mg/mL PMSF. Cell lysis was performed by dounce homogenization, followed by solubilization with 1% (v/v) DDM, 0.2% (w/v) CHS, for 1 hour at 4 °C. The supernatant was collected after high-speed centrifugation at $150,000 \times g$ for 30 min and incubated with

pre-equilibrated ANTI-FLAG M2-Affinity resin (Sigma) for 1 hour. The resin was washed with washing buffer containing 150 mM NaCl, 40 mM HEPES pH 7.4, 0.03% DDM, and 0.006% CHS. ALG10 was eluted with 0.2 mg/mL Flag peptide (GenScript) in washing buffer for 1 hour and concentrated in a 100 KDa-cutoff spin concentrator (Amicon) to a concentration of 1 mg/mL.

### Chemo-enzymatic preparation of Dol20-PP-GlcNAc$_2$Man$_9$Glc$_3$

Enzymatic elongation of Dol20-PP-GlcNAc$_2$ to Dol20-PP-GlcNAc$_2$Man$_9$Glc, from chemically synthesized Dol20-PP-GlcNAc$_2$, Dol25-P-Man and Dol25-P-Glc and commercially available GDP-Man was performed using purified ALG1, ALG2, ALG11, ALG3, ALG9, ALG12, and ALG6 proteins, as previously described[38,39]. The reaction mixture for the addition of the last two glucose contained 150 μM Dol20-PP-GlcNAc$_2$Man$_9$Glc, 500 μM Dol25-P-Glc, 1.2 μM purified ALG8 and 1.2 μM purified ALG10 in a buffer containing 25 mM HEPES, pH 7.5, 150 mM NaCl, 10 mM MgCl$_2$, 0.03% DDM and 0.006% CHS layered with argon. The reaction was incubated overnight at 10 °C and the products were analyzed by in vitro glycosylation of the fluorescently labeled peptide TAMRA-YANATS by purified OST complex, followed by Tricine SDS-PAGE consisting of a 16% resolving gel with 6 M urea, a 10% spacer gel, and a 4% stacking gel[53,54] and LC-MS/MS analysis.

Purification of Dol20-PP-GlcNAc$_2$Man$_9$Glc$_3$ was performed by lyophilization of the reaction mixture, followed by extraction with CHCl$_3$:MeOH (2:1, v/v) to remove contaminants carrying short glycan moieties. A final extraction was performed with CHCl$_3$:MeOH:H$_2$O (10:10:3, v/v/v), and the supernatant was dried under argon and resuspended in a buffer containing 25 mM HEPES pH 7.5 and 150 mM NaCl.

### In vitro glycosylation assays

For evaluation of LLO elongation, reaction mixtures contained 1 μM purified OST, 10 μM LLO analog, 10 mM MnCl$_2$, and 15 μM of the fluorescently labeled peptide 5-TAMRA-YANATS (Genscript) in a final volume of 20 μL, and were incubated at 30 °C for 20 min. The reaction mixtures were either analyzed by mass spectrometry or diluted with SDS Laemmli buffer and analyzed by tricine SDS-PAGE consisting of a 16% resolving gel with 6 M urea, a 10% spacer gel and a 4% stacking gel[53]. Fluorescent bands for peptide and glycopeptide were visualized by using a Sapphire™ biomolecular imager (Azzure).

For kinetic analysis, reaction mixtures contained 0.1 μM purified OST, 30 μM LLO analog, 10 mM MnCl$_2$, and 10 μM of the fluorescently labeled peptide 5-TAMRA-YANATS (Genscript) in a final volume of 20 μL. Reactions were incubated at 30 °C. 2 μL samples were taken at different time points (2, 5 10, 15, 20 30, and 60 min) and diluted in 10% ACN, 10 mM phosphate buffer pH 7. Samples were analyzed by reverse-phase chromatography using a UPLC Dionex UltiMate 3000 with an Accucore 150-C18 100 × 2.1 mm 2.6 μm column (Thermo Fisher Scientific)[40]. Peaks for glycopeptide and peptide were integrated using the Software Chromeleon™, and the amount of produced glycopeptide was determined for each data point. Turnover rates were calculated by fitting the data to linear regression using PRISM software.

### Grid preparation and cryo-EM data acquisition

Cryo-EM grids were prepared using a Vitrobot Mark IV (FEI) with an environmental chamber set at 95% humidity and 4 °C. For the preparation of the ternary-complex sample, nanodiscs-reconstituted OST at a protein concentration of 2.5 μM was incubated with 7.5 μM Dol20-PP-GlcNAc$_2$Man$_9$Glc$_3$, and 500 μM TAMRA-YA(Dab)ATS peptide for 15 min prior to vitrification. For the peptide-bound sample, digitonin-purified OST was concentrated to 18 μM and incubated with 500 μM of TAMRA-YANATS peptide for 15 min prior to vitrification.

Aliquots of 4 μL of sample were placed onto glow-discharged Quantifoil carbon grids (R1.2/1.3, copper, 300 mesh, for ternary-complex; 400 mesh for peptide-bound). Grids were blotted with filter paper for 2.5–3.5 s and flash-frozen in a mixture of liquid ethane and propane cooled by liquid nitrogen. Grids were imaged with a Titan Krios (FEI) electron microscope operated at 300 keV, equipped with a Gatan K2 (ternary-complex) or K3 (peptide-bound), summit direct electron detector and Gatan Imaging Filter, with a slit width of 20 eV to remove inelastically scattered electrons. Movies were recorded semi-automatically with EPU software (Thermo Fisher Scientific) with a defocus range between −0.6 and −2.4 μm and a super-resolution pixel size of 0.42 Å/pixel for ternary-complex, and 0.33 Å/pixel for peptide-bound.

### Cryo-EM data processing and model building

Detailed pipelines of cryo-EM data processing are shown in supplementary figures 2 and 8. In brief, for the ternary-complex sample, two datasets were collected. Movies were corrected for beam-induced motion using MotionCor2[55]. Contrast transfer function (CTF) parameters were estimated using Gctf[56]. Particle-picking was performed in RELION 3.1, using reference-free auto-picking with the Laplacian-of-Gaussian filter, and binned by a factor of 2 during particle extraction. Several rounds of 2D classification were performed and the best classes were selected for one round of 3D classification using the previously published map of yeast OST in nanodisc as a reference. Particles in the best class were selected, re-extracted to 0.85 Å/pixel, and subjected to two rounds of heterogeneous refinement in CryoSPARC 3.2. Good classes were combined and further refined using non-uniform refinement and local refinement yielding a 3D reconstruction at 3.1 Å resolution. 3D variability analysis (3DVA) was performed with particles contributing to this volume. 3DVA outputs were visualized with 3D Variability Display tool in CryoSPARC 3.2. Two main classes, LLO-bound and ternary-complex, were identified and independently refined using non-uniform refinement and local refinement, resulting in 3D volume of 3.0 Å resolution for LLO-bound state and 3.1 Å resolution for ternary-complex.

For the peptide-bound sample one dataset was collected and cryo-EM data processing was performed as described for the ternary-complex. After 3D heterogeneous refinement in CryoSPARC 3.2, one good class was further refined using non-uniform refinement and local refinement yielding a 3D reconstruction at 2.7 Å resolution. A 3DVA was performed with the particles that contributed to this volume. 3DVA outputs were visualized with 3D Variability Display tool in cryosparc 3.2. Two main classes, peptide-bound and apo-state, were identified and independently refined using non-uniform refinement and local refinement, resulting in 3D volume of 2.8 Å resolution for peptide-bound state and 2.9 Å resolution for the apo-state.

Model building was performed in Coot 0.9[57], using the final maps shown in supplementary figs. 2 and 8, and the apo-OST model PDB ID:6EZN[28]. The electron density was of excellent quality for the bound substrates and allowed for the unambiguous building of the model. The glycan moiety of the LLO was built using the carbohydrate module in Coot 0.9. Final models were refined in PHENIX version 1.17.1[58].

### Figure preparation and data analysis

Structural figures were prepared using PyMol 4.6.0 and UCSF ChimeraX 0.9[59]. Kinetic graphs and analysis were performed using GraphPad Prism 9.0, GraphPad Software, San Diego, California USA, www.graphpad.com. Protein sequence alignments were generated by EMBOSS[60] and were prepared for visualization using JalView 2.11.0[61].

### Reporting summary

Further information on research design is available in the Nature Portfolio Reporting Summary linked to this article.

## Data availability

Atomic coordinates of the models were deposited in the RCSB Protein Data Bank (PDB) under accession number 8AGB for OST-LLO-primed,

8AGC for OST-ternary-complex, and 8AGE for OST-peptide-primed. The three-dimensional cryo-EM maps were deposited in the Electron Microscopy Data Bank (EMDB) under accession numbers EMD-15419 for OST-LLO-primed, EMD-15420 for OST-ternary-complex and EMD-15421 for OST-peptide-primed. Atomic coordinates of the apo structure of OST accession number 6EZN were used as initial model. Source data are provided with this paper. All other data are available from the corresponding author upon reasonable request. Source data are provided with this paper.

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

## Acknowledgements

We thank the staff at the Scientific Center for Optical and Electron Microscopy (ScopeM, ETH Zurich, Switzerland) for support during cryo-EM data collection; Dr. Chia-wei Lin and the functional genomic center Zurich (FGCZ) for mass spectrometry analysis; Anna-Lena Schinke and Meike Mikolin for help with protein expression and cell culture work; Tamis Darbre for helpful discussions; Andrew Alexander for critical reading of the manuscript. This work was supported by the Swiss National Science Foundation (grant CRSII5_173709 to J.-L.R., M.A., and K.P.L. and grant 310030_196862 to K.P.L and J.-L.R.).

## Author contributions

A.S.R. and K.P.L conceived the project and designed the experiments. A.S.R. performed expression, purification, and biochemical character-ization of yeast OST complex. M.d.C. and G.P. performed the chemical synthesis of the LLO precursor Dol20-PP-GlcNAc$_2$ and the donor sub-strates Dol25-P-Man and Dol25-P-Glc. A.S.R. produced recombinant ALG1, ALG2, and ALG11 enzymes and established the production of human ALG8. J.S.B. produced recombinant ALG3, ALG9, ALG12, and ALG6 enzymes and established the production of yeast ALG10. A.S.R. and J.S.B. performed the chemo-enzymatic synthesis of Dol20-PP-GlcNAc$_2$Man$_9$Glc$_3$. J.K. and R.N.I. collected cryo-EM data. A.S.R. and J.K. prepared cryo-EM grids and processed cryo-EM data. A.S.R. built, refined, and validate the OST models with help from K.P.L. J.-L.R. supervised the synthesis of LLO substrates. A.S.R., M.d.C., G.P., J.K., J.-L.R., M.A., and K.P.L. analyzed the data. A.S.R., M.A., and K.P.L. wrote the paper with the input of all authors.

## Competing interests
