## [Peer Review File · Nature Communications]

Molecular basis for glycan recognition and reaction priming of eukaryotic oligosaccharyltransferaseReviewer #1 (Remarks to the Author):

In this work, Ramírez and co-workers reveal the structural basis of glycan recognition by eukaryotic oligosaccharyltransferases (OST). OST is the central enzyme of N-linked protein glycosylation, an essential process in eukaryotes. It catalyzes the transfer of a pre-assembled glycan, GlcNAc2Man9Glc3, from a dolichyl-pyrophosphate donor to acceptor sites in secretory proteins in the lumen of the endoplasmic reticulum (ER).

Specifically, the authors present three cryo-EM structures of yeast OST: (i) a binary complex with the lipid sugar donor chemo-enzymatically synthesized Dol20-PP-GlcNAc2Man9Glc3, at 3.0Å (LLO-primed), (ii) a binary complex with the acceptor reactive peptide TAMRA-YANATS-NH₂, at 2.8Å (peptide-primed), and (iii) a ternary complex with Dol20-PP-GlcNAc2Man9Glc3 and TAMRA-YA(Dab)ATS-NH₂, at 3.1Å. Yeast OST is a multimeric complex composed of eight subunits spatially arranged in three subcomplexes that are formed during OST assembly. In the structure containing Dol20-PP-GlcNAc2Man9Glc3, the authors observed well-defined EM density covering the divalent metal cation Mn²⁺, the four isoprenyl units of Dol20, the pyrophosphate group, and the entire A-branch glycan, which includes two GlcNAc units at the reducing end, four mannoses, and the three terminal glucoses. Interestingly, the A-branch of the glycan revealed an extended conformation, oriented parallel to the membrane plane. The recognition of GlcNAc2 exclusively involves residues of the catalytic subunit STT3. At the non-reducing end, the terminal Glc are accommodated in a binding pocket formed by the auxiliary subunits OST2 and WBP1. In the ternary complex, the external loop EL5 of the STT3 subunit is ordered and packs against the luminal domain of STT3, resulting in the formation of a tunnel that connects the binding sites of acceptor and donor substrates. The structural and enzymatic data support a mechanism for the OST-catalyzed reaction cycle, which involves structural rearrangements of the EL5 and TM9 regions of the catalytic subunit STT3.

Finally, the authors analyzed WBP1 and OST2 residues involved in forming the binding site of the Glc3 moiety and found a high conservation of the residues interacting with the terminal α ,1-2 glucoses in multimeric OSTs that transfer Glc3-containing oligosaccharides. These analyses suggest a coordinated evolutionary path of N-linked protein glycosylation reflected in OST architecture and oligosaccharide structure. The article is clearly and well written and the interpretation of the experimental data support the proposed model. Because of its novelty and the quality of the work, I strongly support the publication of this manuscript in Nature Communications.

Some suggestions for improvement are listed below:

1. page 25: Figure 4c. I suggest adding close-up views for each 8-OST, 7-OST, 6-OST and 4-OST – surface and cartoon representations – showing (i) the conservation of the GlcNAc2 binding site and (ii) the presence/absence of the Glc3 binding pocket.

2. page 25: Figure 4c. Could the authors kindly add some information on the Trypanosoma OST systems?

3. Spelling:

- Please, use/homogenize Glc, Man, and GlcNAc as abbreviations for glucose, mannose, and N-acetylglucosamine throughout the text.

Congratulations to the authors for this very nice work.

Reviewer #2 (Remarks to the Author):

Transfer of 14-sugar glycans from LLOs to Asn residues of secretory pathway cargo is widely important for their folding and their eventual functions. This structural study is a seminal contribution that, for the first time, reveals a detailed mechanism for the critical selectivity of most

eukaryotic OST complexes for tri-glucosylated donor LLOs. It is these glucose residues that optimize OST activity, and orchestrate initial steps in folding and quality control. The study also reveals that instead of binding the 14-sugar glycan along its body by a typical lectin-like mechanism as many had expected, it is a novel mechanism more like a guitar string strongly tethered by the sugars at each end, with the center sugars held rigid but without specific binding interaction. Surprisingly, the tri-glucosyl end is held in place by a novel pocket formed by two accessory subunits, WBP1 and OST2, rather than the major core catalytic subunit STT3.

They convincingly propose a catalytic mechanism in which "the binding of either donor or acceptor substrate to OST leads to a primed state of the enzyme, where subsequent binding of the other substrate triggers conformational changes required for catalysis". Their chemoenzymatic production of pure 14-sugar LLOs for these studies is highly impressive -- they will no doubt receive many requests for this reagent!

Overall, this paper is an elegant capstone to a series of important cryo-EM studies over the past years from this and other groups, seeking to understand the detailed mechanism and logic of the OST enzyme complex. OST has been widely studied since the 1970s. Thanks to this paper, I now feel that I have a comprehensive understanding of how OST works.

This reviewer has only MINOR points for the authors to consider and address at their discretion.

1. In some key places early in the paper the OST is referred to as "yeast OST" without noting the strain. I assume *S. cerevisiae*.
2. would this be more appropriate? "... the sequential action of multiple glycosyltransferases, [named] encoded by ALG (Asparagine-Linked-Glycosylation) [enzymes] genes." I say this because Alg13 and Alg14, for example, are two genes but comprise 1 enzyme.
3. "...ER-luminal lectins calreticulin (CRT) and calnexin (CNX), which recruit specific chaperones to direct the glycoproteins either to folding or to degradation..."
In this paragraph would it be appropriate to also mention other glycan intermediates and relevant binding partners such as malectin, Unfolded Glycoprotein Glucosyltransferase, and OS9?
4. Would this change be better? "This leads to completion of [incomplete LLO intermediates and] fully assembled DolPP-GlcNAc2Man9Glc3 as donor substrates for OST3"
5. "The binding of donor substrate Dol20-PP-GlcNAc2Man9Glc3 alone did not alter the structure of STT3. In particular, EL5 and TM9 remained disordered (Figure 2A) and the active site of STT3 is not fully formed, which probably helps preventing futile hydrolysis of bound LLO substrate." Compared with the new data here on the yeast OST complex, please comment on ability of mammalian STT3B-OST to hydrolyze LLO in the absence of acceptor peptide (<https://doi.org/10.1073/pnas.1806034115> and references therein):
6. I think the analysis presented in Fig 4 in the Discussion is new information with important insights for the field, and addresses a question that is prompted in reader's mind by the preceding data figures. Would it be more appropriate to include Fig 4 in the Results?
7. For the OST enzymatic assays in Figs 1B, 1C, 2D, I don't see the minutes of incubation in the Legends or the Methods section. If the incubations are short, I would infer that they emulate an initial rate. If they are long, then factors such as stabilization by substrate binding may come into play.
8. I think in the earlier literature EL5 was proposed to be important for securing the glycan in a fairly broad pocket. Now, it actually seems more important for forming the active site (Fig 5). Do I have that correct?

Reviewer response letter of NCOMMS-22-40156-T

We thank the reviewers for their insightful and constructive comments. In the following, reviewer comments are in italics and our responses / actions are in plain text colored red.

Reviewer 1:

Some suggestions for improvement are listed below:

1. page 25: Figure 4c. I suggest adding close-up views for each 8-OST, 7-OST, 6-OST and 4-OST – surface and cartoon representations – showing (i) the conservation of the GlcNAc2 binding site and (ii) the presence/absence of the Glc3 binding pocket.

Given that the models shown for 7-OST, 6-OST and 4-OST are not experimental structures but models generated by removing subunits from the 8-OST, we think that the close-up views would be speculative and not provide additional experimental information.

2. page 25: Figure 4c. Could the authors kindly add some information on the Trypanosoma OST systems?

We did not include *Trypanosoma* OST systems in Fig. 4c because they are single-subunit OSTs (no additional subunits that could provide glycan binding). However, we have added STT3A from *Trypanosoma brucei* in the protein sequence alignment in Fig. 4b.

3. Spelling:

- Please, use/homogenize Glc, Man, and GlcNAc as abbreviations for glucose, mannose, and N-acetylglucosamine throughout the text.

We have corrected this throughout the manuscript.

Congratulations to the authors for this very nice work.

Reviewer #2:

This reviewer has only MINOR points for the authors to consider and address at their discretion.

1. In some key placed early in the paper the OST is referred to as “yeast OST” without noting the strain. I assume S. cerevisiae.

We modified the following sentence in the last paragraph of the introduction, as shown below (Page 3):

“Here we present high-resolution structures of **S. cerevisiae OST (yeast OST)** bound to chemically well-defined substrate analogs....”

2. would this be more appropriate? "... the sequential action of multiple glycosyltransferases, [named] encoded by ALG (Asparagine-Linked-Glycosylation) [enzymes] genes." I say this because Alg13 and Alg14, for example, are two genes but comprise 1 enzyme.

We agree and have modified the text accordingly (Page 3)

3. "...ER-luminal lectins calreticulin (CRT) and calnexin (CNX), which recruit specific chaperones to direct the glycoproteins either to folding or to degradation..."
In this paragraph would it be appropriate to also mention other glycan intermediates and relevant binding partners such as malectin, Unfolded Glycoprotein Glucosyltransferase, and OS9?

Although we acknowledge the importance of N-glycan trimming and the CRT/CNX glycoprotein folding pathway, we think that a more detailed description might not be needed, given that our study focuses on the recognition of the fully-assembled oligosaccharide before OST-catalyzed transfer.

4. Would this change be better? "This leads to completion of [incomplete LLO intermediates and] fully assembled DoIPP-GlcNAc₂Man₉Glc₃ as donor substrates for OST3"

We think that in the original sentence "This leads to a competition of incomplete LLO intermediates and fully assembled DoIPP-GlcNAc₂Man₉Glc₃ as donor substrates for OST" the expression "LLO intermediates" already indicates they are incomplete. Therefore, we preferred to keep the original sentence (Page 3).

5. "The binding of donor substrate DoI20-PP-GlcNAc₂Man₉Glc₃ alone did not alter the structure of STT3. In particular, EL5 and TM9 remained disordered (Figure 2A) and the active site of STT3 is not fully formed, which probably helps preventing futile hydrolysis of bound LLO substrate."

Compared with the new data here on the yeast OST complex, please comment on ability of mammalian STT3B-OST to hydrolyze LLO in the absence of acceptor peptide (<https://doi.org/10.1073/pnas.1806034115> and references therein):

The hydrolysis of LLO molecules by OST-B has been shown to be ~100 fold less efficient than the glycan transfer reaction (Lu, et al, <https://doi.org/10.1073/pnas.1806034115>). We think that discussing the ability of OST-B to hydrolyze LLO molecules *in vivo* based on our structures would be too speculative.

6. I think the analysis presented in Fig 4 in the Discussion is new information with important insights for the field, and addresses a question that is prompted in reader's mind by the preceding data figures. Would it be more appropriate to include Fig 4 in the Results?

We agree with this suggestion and modified/rearranged the text accordingly (Pages 6-9).

7. For the OST enzymatic assays in Figs 1B, 1C, 2D, I don't see the minutes of incubation in the Legends or the Methods section. If the incubations are short, I would infer that they emulate an initial rate. If they are long, then factors such as stabilization by substrate binding may come into play.

This information is now included in the Methods section (Page 12, 13).

8. *I think in the earlier literature EL5 was proposed to be important for securing the glycan in a fairly broad pocket. Now, it actually seems more important for forming the active site (Fig 5). Do I have that correct?*

Yes. The role of EL5 in completing the active site of OST enzymes seems to be conserved in the different OST systems. Regarding EL5 role in LLO binding, earlier studies on the bacterial OST PglB, proposed that binding of LLO requires the N-terminal half of EL5 (N-EL5) to be disordered, and that in an LLO-bound state N-EL5 might be engaged. Our results in the eukaryotic system, revealed that EL5 is not engaged upon LLO binding alone, but only after simultaneous binding of both substrates.